# *In Situ* Visual Distribution of Gelsemine, Koumine, and Gelsenicine by MSI in *Gelsemium*
*elegans* at Different Growth Stages

**DOI:** 10.3390/molecules27061810

**Published:** 2022-03-10

**Authors:** Zi-Han Wu, Yi Su, Zhou-Fei Luo, Zhi-Liang Sun, Zhi-Hong Gong, Lang-Tao Xiao

**Affiliations:** 1College of Bioscience and Biotechnology, Hunan Agricultural University, Changsha 410128, China; wuzihan@stu.hunau.edu.cn (Z.-H.W.); yisu@hunau.edu.cn (Y.S.); zhoufeiluo@hunau.edu.cn (Z.-F.L.); 2College of Veterinary Medicine, Hunan Agricultural University, Changsha 410128, China; 3Waters Technology (Shanghai) Co., Ltd., Shanghai 200120, China; zhihong_gong@waters.com

**Keywords:** *G. elegans*, DESI-MSI, distribution, alkaloid, visualization, LC-MS/MS

## Abstract

The distribution of pharmatically important alkaloids gelsemine, koumine, and gelsenicine in *Gelsemium elegans* tissues is a hot topic attracting research attention. Regretfully, the in planta visual distribution details of these alkaloids are far from clear although several researches reported the alkaloid quantification in *G. elegans* by LC-MS/MS. In this study, mass imaging spectrometry (MSI) was employed to visualize the *in situ* visualization of gelsemine, koumine, and gelsenicine in different organs and tissues of *G. elegans* at different growth stages, and the relative quantification of three alkaloids were performed according to the image brightness intensities captured by the desorption electrospray ionization MSI (DESI-MSI). The results indicated that these alkaloids were mainly accumulated in pith region and gradually decreased from pith to epidermis. Interestingly, three alkaloids were found to be present in higher abundance in the leaf vein. Along with the growth and development, the accumulation of these alkaloids was gradually increased in root and stem. Moreover, we employed LC-MS/MS to quantify three alkaloids and further validated the *in situ* distributions. The content of koumine reached 249.2 μg/g in mature roots, 272.0 μg/g in mature leaves, and 149.1 μg/g in mature stems, respectively, which is significantly higher than that of gelsemine and gelsenicine in the same organ. This study provided an accurately *in situ* visualization of gelsemine, koumine, and gelsenicine in *G. elegans*, and would be helpful for understanding their accumulation in plant and guiding application.

## 1. Introduction

*Gelsemium* belongs to the Gelsemicaeae family [1]. Among the three species identified in *Gelsemium*, *Gelsemium elegans* is distributed in Asia, *Gelsemium sempervirens* and *Gelsemium rankinii* are distributed in North America [2]. Morphological characteristics analysis showed that *Gelsemium sempervirens* and *Gelsemium rankinii* are sister taxa [3]. *Gelsemium elegans* is widely distributed in southern China [4], it grows well at high altitude area and that makes the plant popular among hilltribes. *G. elegans* shows relatively high toxicity due to its neurological and respiratory depressive effects. It can lead to violent clonic convulsions and respiratory failure at a lethal dose [5]. In spite of its potential toxicity, *G. elegans* shows high medicinal value and is regarded as an important medicinal plant in China. *G. elegans* has strong pharmacological and pharmacodynamic effects on human and animals since it can be used to reduce anxiety, to manage pain and to inhibit inflammation [6,7,8,9,10]. More than 200 compounds with drug activity, including alkaloids, iridoids, and steroids, have been identified in *Gelsemium*. Some alkaloids and iridoids exist only in the *Gelsemium* genus, and are regarded as the bio-active compounds responsible for the observed pharmacological effects [11,12]. Currently, at least 149 alkaloids with a basic structure of indole, oxindole or bisindole nuclei have been isolated and identified in *Gelsemium* [13]. According to the chemical skeleton, alkaloids are classified into six groups: gelsemine-type, gelsedine-type, sarpagine-type, humantenine-type, koumine-type, and yohimbane-type [2,14,15].

Based on the pharmacological and pharmacodynamic effects, scientists paid more attentions on gelsemine, koumine, and gelsenicine in recent years. Gelsemine isolated in *Gelsemium* showed anxiolytic and neuralgia alleviating effects, and can be used to treat nervous system diseases [16]. Gelsemine exerts its anxiolytic action through stimulating the local production of neurosteroid allopregnanolone to result in positive allosteric activation of GABAA receptors [17]. Koumine, which was isolated in *G. elegans* in 1931 and was structurally identified in 1981 [18], was highly cytotoxic to cancer cell, and showed a skin disease inhibition and analgesic activity but no addictive effects. In addition, koumine can increase allopregnanolone in the spinal cord, thus showed an analgesic action for inflammatory and neuropathic pain [19]. The efficacy of gelsenicine was similar to koumine [20], but the therapeutic index of koumine is wider than gelsenicine [19]. 

For better utilization of herbal medicine, the distribution of medicinal compositions in plant tissues/organs is paid more attention. Through HPLC, QqTOF or MSn, previous studies roughly revealed the in planta distribution in *G. elegans* [21,22,23,24,25,26], e.g., higher gelsenicine in roots and branches [27,28] and more gelsemine in roots, stems and leaves [29,30]. Laser induced light backscattering imaging, stained sample microscopic observation and isotope tracer technique were also employed to investigate the distribution of analytes [31]. In addition, a new photoluminescence sensor based on carbon dot and nanocomposites [32,33,34] also showed application potential in the determination of active ingredients in medicinal plant. In recent years, mass imaging spectrometry (MSI) has emerged as a novel label-free and sensitive imaging technology. MSI combines the multichannel (*m*/*z*) measurement capability of mass spectrometer and imaging technology, showing great potentials in *in situ* distribution biological molecules in tissues [35,36,37,38]. MSI was first applied in medical and pharmaceutical research. Among the MSI methods, matrix-assisted laser desorption/ionization MSI (MALDI-MSI) has been applied to locate some functional components in plants [39,40,41], but its limited quality of the matrix deposit on the samples could result in decreased resolution and misinterpretation. Comparatively, desorption electrospray ionization MSI (DESI-MSI) can display a clearer localization of metabolites, and shows greater advantages in *in situ* imaging.

Along with the increasing attention of herbal medicine, researchers and pharmaceutists are focusing on pharmacodynamic components and their fine spatiotemporal distribution in medicinal plants. Among them, gelsemine, koumine, and gelsenicine were three bio-active alkaloids extracted from *G. elegans* which have important pharmacological functions. The contents of these alkaloids in plants were well quantified through LC-MS/MS, and several studies preliminarily revealed the rough distributions of these alkaloids in *G. elegans*. However, *in situ* distributions of gelsemine, koumine, and gelsenicine in *G. elegans* are far from clear, and their exact spatiotemporal accumulation cannot be visualized in plant organs/tissues because of the insurmountable limitation of traditional chromatography and mass spectrometry methods. Therefore, DESI-MSI was employed to image the dynamic accumulations of gelsemine, koumine, gelsenicine in different organs/tissues during the growth and development of *G. elegans* in this study. Furthermore, the *in situ* distribution of three alkaloids were validated through LC-MS/MS quantification. Our studies revealed a high-resolution *in situ* distribution map of the three alkaloids in *G. elegans* and provided a useful protocol for the visualized *in situ* quantitative distribution of alkaloids in plants. 

## 2. Results

### 2.1. In Situ Visualization of Three Alkaloids in Mature Organs/Tissues

*G. elegans* is an evergreen vine species that is widely distributed in southern subtropical regions of China. *G. elegans* has smooth and twining stems that contains milky latex, and its leaves are opposite, entire and glabrous. The root system of *G. elegans* is very developed. Previous reports revealed that three important indole alkaloids (gelsemine, koumine, and gelsenicine) (Figure 1a–c) extracted in *G. elegans* (Figure 1d) showed a variety of pharmacological activities [42]. Understanding the spatial distribution of alkaloids is the premise for efficient extraction in *G. elegans*. The analytical methods based on UPLC, LC-MS/MS, and GC-MS/MS are unable to achieve *in situ* and visual analysis for the compounds in *G. elegans* tissues. In this study, DESI-MSI was employed to visualize the spatial distribution of gelsemine, koumine, and gelsenicine in frozen sections derived from different *G. elegans* organs/tissues. In order to validate the DESI-MS imaging based *in situ* distribution of three alkaloids in *G. elegans*, HPLC-MS/MS quantification was performed by detecting their precise contents in different tissues/organs. 

The distributions of gelsemine, koumine, and gelsenicine in various parts were displayed in Figure 1b. The ion mass spectra were shown in Appendix A. In roots, gelsemine, koumine, and gelsenicine were mainly located in vascular bundle region and were decreased gradually from pith to epidermis. Compared to that in root, three alkaloids in stem displayed a similar localization pattern, i.e., the three alkaloids were mainly located in the pith region of stem, and were significantly decreased in the epidermis. Interestingly, gelsemine, koumine, and gelsenicine were present in higher abundance in the leaf vein. The DESI-MSI results were consistent with the quantification by using LC-MS/MS (Figure 1e–g and Appendix A). The leaves contained a higher content of alkaloids, the contents of gelsemine, koumine, gelsenicine in leaves were respectively 122.4 μg/g, 272.0 μg/g, 155.1 μg/g. Additionally, the content of koumine was much higher than that of gelsemine and gelsenicine wherever in roots, stems, or leaves, respectively reached to 249.2 μg/g, 149.1 μg/g, and 272.0 μg/g. The results indicated that the content of koumine was much higher than that of gelsemine and gelsenicine wherever in roots, stems or leaves, but the contents of gelsemine and gelsenicine were very close to each other.

### 2.2. Visualization of Three Alkaloids in G. elegans Seedlings at Different Stages

In order to discover the distribution dynamics of three alkaloids during the growth and development, 30 day, 60 day, and 90 day old *G. elegans* seedlings were used as the materials, three alkaloids in different plant organs/tissues at different growth stages were visualized by DESI-MSI (Figure 2a–c). Along with the plant growth, alkaloids gradually accumulated in plant tissues. The accumulations of three alkaloids were started in the endodermis/pith region of root and stem (Figure 3d–f). From 30 day to 60 day, gelsemine, koumine, and gelsenicine showed a stably slow increase along with the growth and development. The three alkaloids were still detected mainly in vascular bundle (especially in pith) region of root and stem (Figure 3d–f). Meanwhile, the accumulation rate of the three alkaloids is significantly different. Less accumulation of gelsenicine and gelsemine was found in root, stem and leaf, respectively. At 90 day, all three alkaloids showed high accumulation. The results indicated that three alkaloid contents in *G. elegans* tissue are relevant to the plant age.

### 2.3. Relative Quantification of Three Alkaloids Based on Image Brightness Intensities

The map screened by DESI-MSI revealed the *in situ* distribution of three alkaloids and the image brightness intensities represented their relative contents in a frozen section (Figure 1b and Figure 2d–f). In fact, the brightness in the map was uneven and same brightness area was also irregular. In order to better understand the distribution, the area percentages (AP), which is the ratio of the area of a certain intensity range to the total area of tissue/organ section, was employed to relatively quantify the content of three alkaloids. According to Figure 1b, the image brightness intensities in frozen section were divided into three ranges: >1400, 700–1400, and <700, and AP of each alkaloid in the three ranges were evaluated in different tissue slices (Table 1). The gelsenicine content was very low in roots (AP was more than 96% in intensity range <700), but was very high in stem, and leaf (APs were 100%, 72.4%, and 72.5%, respectively, in intensity range >1400). The koumine content was very high in four tissues (APs were 62.4%, 100%, 78.9%, and 83.6%, respectively, in intensity range >1700). The gelsemine content was low in leaf (APs were 4.9% in intensity range >1700). The results were consistent with the quantification by using LC-MS/MS (Figure 1c–e).

According to the image brightness intensities in Figure 2d–f, the relative contents of three alkaloids were evaluated in different organs/tissues. The image brightness intensities of root, stem and leave were respectively divided into three ranges (roots: >1400, 700–1400, and <700; stems: >5300, 2600–5300, and <2600; leaves: >2300, 1100–2300, and <1100), AP in the three ranges were evaluated in different organs/tissues (Figure 3). The three alkaloids contents in stems were generally higher than in roots and leaves. The AP of gelsemine, koumine, and gelsenicine in intensity range >1400, >5300, and >2300 showed increase trends, which indicated that three alkaloids continuous produced and accumulated during the growth of seedling. The gelsenicine content was very low in 30 day root (APs were 0% in intensity range >1400, 6.6% in intensity range 700–1400 and 93.4% in intensity range <700), but a significant increase was detected at 60 day and 90 day (APs were 30.5% and 40.7%, respectively, in intensity range >1400). In 30 day and 60 day leaves, the distribution of gelsemine, koumine, and gelsenicine did not concentrate in the veins, but in 90 day leaves, three alkaloids began to concentrate mainly in the veins, the results were same as the previous results of the mature leaves (Figure 1b).

### 2.4. Method Validation

For confirming the results by MSI, the precise contents of gelsemine, koumine, and gelsenicine were determined in corresponding tissues/organs by using LC-MS/MS. Firstly, the linearity, limit of detection (LOD) and limit of quantification (LOQ) were investigated to validate the LC-MS/MS method. Calibration samples at five concentrations (10, 50, 100, 200, and 500 ng/mL) were used to generate standard curve (Appendix A). The correlation coefficients of the three compounds were higher than 0.99 in the concentration range of 10–500 ng/mL. The LOD and LOQ of gelsemine were 5 ng/mL and 10 ng/mL respectively. The LOD and LOQ of koumine were 2.5 ng/mL and 10 ng/mL, respectively. The LOD and LOQ of gelsenicine were 2.5 ng/mL and 10 ng/mL, respectively. Then the contents were calculated. It was confirmed that the results of LC-MS/MS quantification was consistent with that of DESI-MSI visualization. The contents of koumine were relatively high whether in the roots, stems, leaves, or at different growth stages (Figure 4a–c). The gelsenicine concentration at 90 day was significantly higher than at 60 day (*p* < 0.001). From 30 day to 60 day to 90 day, the contents of gelsenicine increased from 32.9 μg/g to 46.2 μg/g to 104.1 μg/g. *G. elegans* root contained a relatively high level of koumine. The content of koumine at 30 day root was 53.1 μg/g and sharply increased to 116.4 μg/g in 90 day root (Figure 4a–c). These results also indicated that the accumulation of gelsemine, koumine, and gelsenicine gradually increased along with the plant growth and development. Therefore, maturity might be an index reflecting their accumulation. 

Since organs/tissues of different maturity often coexist in a plant at the same time, *G. elegans* plant was divided into more parts according to the maturity. Young, mature, and senescent leaves were collected, and then the mesophyll and veins were sampled (Figure 4d). The roots were divided into basal root, root tip, and lateral root, and the stems were divided into old, mature, and young stem (Figure 4e–g). Then, the contents of three alkaloids were quantified by using HPLC-MS/MS. The results showed that the contents of three alkaloids were not significantly different between mesophyll and vein in young leaf. The contents of three alkaloids in mature and senescent leaf were higher than that in young leaf (Figure 4h–j). In mature leaves, the contents of gelsemine, koumine, and gelsenicine in leaf vein were significantly higher than that in mesophyll, and the ratios reached 3.2, 1.84, and 2.08, respectively. The results indicated that three alkaloids were mainly gathered in leaf vein, which were consistent with that detected by MSI in mature *G. elegans* leaves (Figure 1). In addition, the significant decrease of alkaloids in senescent leaf indicated that alkaloids in leaf would be degraded along with the aging of plants. Gelsemine, koumine, and gelsenicine were mostly enriched in old root and least in root hair (Figure 4k–m). Three alkaloids presented different distribution in different parts of stems. The contents of koumine and gelsenicine gradually increased from the old stem to the young stem, but gelsemine showed an opposite distribution (Figure 4n–p).

## 3. Discussion

At present, a hot topic in medicinal plant research is identification and quantification of medicinal compounds in plant organ/tissue. For better application of medicinal plant and its active ingredients, pharmaceutists are eager to visualize the detailed *in situ* distribution and accumulation of these compounds in medicinal plant. As main steam methods for metabolite analysis, chromatography and mass spectrometry are powerful tools for identification and quantification of analytes [43,44]; however, these methods are unable to visualize the analytes. Interestingly, the emerging MSI technology is attracting more attention because of its great potential in accurately visualizing the spatial distribution of secondary metabolites in plant tissues [45,46]. MSI was employed to visualize the cell-specific localization of vinblastine and vincristine in *Catharanthus roseus* [47]. MALDI-MS approach was developed to visualize vinca alkaloids in plant petal [48], peptide in mouse brain tissue [49] and disaccharide isomers in onion bulb [50]. DESI-MSI was applied in visualization of several alkaloids in *Rauvolfia tetraphylla* organs [51] and *Catharanthus roseus* petal [52]. Moreover, these researches revealed that the accumulation of alkaloids showed tissue specificity and was largely affected by the external environment. In this study, we employed DESI-MSI to clarify the *in situ* visual distributions of gelsemine, koumine and gelsenicine in *G. elegans* (Figure 1 and Figure 2). Comparing with previous researches, we paid more attentions to the accumulation of these three alkaloids in plant at different growth and development stages. We also divided the plant tissue categories in detail and analyzed the contents in various tissues. Our results indicated that gelsemine, koumine and gelsenicine were located in the pith region of root and decreased gradually from the pith to epidermis. Three alkaloids in stems showed a similar localization pattern comparing to that in root, and presented higher abundance in leaf vein. Furthermore, the diagrammatic map outlined the *in situ* distribution of three alkaloids in *G. elegans* according to the data detected by DESI-MSI and LC-MS/MS (Figure 5). This study provided a spatiotemporal framework of gelsemine, koumine, and gelsenicine in various parts of *G. elegans*, and would contribute to the understanding for precise *in situ* localizations of alkaloids in *G. elegans*. It also would be helpful for exploring the biosynthesis, accumulation and transportation mechanisms of alkaloids.

As an important medicinal plant, *G. elegans* contains abundant alkaloids with monoterpenoid indole structure showing high therapeutic activities. Gelsemine, koumine, and gelsenicine are three representative alkaloids extracted from *G. elegans* which attract more research attentions due to their cytotoxic, analgesic, anxiolytic, anti-inflammatory, and immunomodulating activities [21,42]. Thus, these three alkaloids, which were quantified by HPLC coupled with photo-diode array and quadrupole time-of-flight mass spectrometry (UHPLC-PDA-QTOF/MS), were used as important indexes to assess the quality of *G. elegans* material as medicinal plant [13]. Fingerprint analysis based on the identification of chemical constituents (including gelsemine, koumine, and gelsenicine) was carried out by HPLC coupled with quadrupole-time-of-flight mass spectrometry (QTOF/MS) in *Gelsemium elegans* [53]. In general, previous researches about *G. elegans* mainly focused on identification, separation, and quantification of medicinal ingredients. In this study, we mainly investigated the *in situ* distribution of gelsemine, koumine, and gelsenicine in different organs/tissues at different growth stages by using DESI-MSI and LC-MS/MS, and provided a detailed spatiotemporal distribution of the three alkaloids in *G. elegans* (Figure 1, Figure 2 and Figure 3). More accurate content ranges of these alkaloid were defined in *G. elegans*. e.g., gelsemine, koumine, and gelsenicine were mainly distributed in mature leaf, respectively, ranged from 130.1 to 114.7 μg/g, 282.0 to 262.0 μg/g, and 159.3 to 150.9 μg/g, and *G. elegans* contained higher koumine, especially in mature leaf, and its content reached 282.0–262.0 μg/g. These results would provide reference for the extraction, separation, quantification, and localization of gelsemine, koumine, and gelsenicine, and also some insights for future investigations on the biological functions of these alkaloids in *G. elegans*.

## 4. Materials and Methods

### 4.1. Chemicals and Reagents 

Saccharose was purchased from the Sinopharm Chemical Reagent Co., Ltd. (Shanghai, China). Methanol and acetonitrile were purchased from Merck Chemicals Co. (Darmstadt, Germany). Fixation fluid was obtained from Wuhan Servicebio Technology Co., Ltd. (Wuhan, China). Optimum Cutting Polymer (O.C.T) was purchased from Sakura (Torrance, CA, USA). Ultrapure water (resistivity ≥18.25 MΩ/cm) obtained from WaterPro water system (ULUPURE, Chengdu, China). Standard chemicals of gelsemine (CAS#: 509-15-9, purity: ≥98%), koumine (CAS#: 1358-76-5, purity: ≥99%), and gelsenicine (CAS#: 82354-38-9, purity: ≥99%) were purchased from the Chengdu Must Bio-technology Co., Ltd. (Chengdu, China). All the reagents and chemicals employed in the experiments were of superior analytical grade (least 98% purity).

### 4.2. Plant Growth Conditions

The seeds and plants of *G. elegans* were collected from Longyan city, Fujian province of China (N 24°43′12″, E 116°43′48″). *G. elegans* seeds were soaked in distilled water for three days, and then the germinated seeds were transplanted into peat soil pots (PINDSTRUP, Denmark). A total of 50 pots (each plot contained 2 plants) of plants were planted. The plants were grown in greenhouse under the following conditions: 16/8 h day/night cycle, 24/20 °C day/night temperature, 70% humidity, and 200 μmol m^−2^ s^−1^ light intensity. Plant materials were divided into two parts: one part was used to make frozen sections for alkaloid imaging by DESI-MSI, and the other part was used for quantification by LC-MS/MS.

### 4.3. Frozen Section Preparation for DESI-MSI

The plant tissues/organs of 30, 60, and 90 day old seedling were collected. The roots, stems, and leaves of mature *G. elegans* were also prepared. Samples were cross-sectioned by surgical blade and fixed with the fixative solution for 24 h. The trimmed tissues/organs were placed in 15% sucrose solution and sunk at 4 °C freezer (Haier BCD-315TNGS, Qingdao, China) for dehydration. The tissues/organs were transferred to 30% sucrose solution and sunk at 4 °C freezer for dehydration again. The dewatered tissues/organs were taken out and placed on the embedding machine (Thermo HistoStar, Waltham, MA, USA) with the cut surface facing up. O.C.T. was dripped around the tissues/organs, and the embedding machine was placed on the microtome cryostat (Thermo CRYOSTAR NX50, Waltham, MA, USA) for quick freezing and embedding. The white and hard O.C.T. was sliced into 8–10 μm slices. Three adjacent sections were selected for detection. Glass slides (Wuhan Servicebio Technology Co., Ltd. Wuhan, China) were placed on top of the sample slices to obtain frozen sections. To minimize analyte degradation, the sample slides were stored at −80 °C freezer (Haier DW-86L388J, Qingdao, China) until analyzed by DESI-MSI.

### 4.4. DESI-MSI Instrumentation and Data Acquisition

DESI-QToF-MS (Waters Xevo G2-XS, Milford, MA, USA) was used for alkaloid imaging. The DESI source conditions were as follows: the nebulizing gas (dry nitrogen) pressure was 5 bar; the spray solvent was methanol: water = 98:2 (*v*/*v*) and the flow rate was 2 μL/min; solvent was sprayed onto the sample surface at an angle of 60; the horizontal distance between the tip of spray and the mass spectrometer inlet was 3 mm, spray port to sample surface was 2 mm and the mass spectrometer inlet was 0.5 mm from the sample surface; the data collection was in positive ion mode, the scanning mass range was *m*/*z* 50–1200, the spray voltage was 4.50 kV and the ion source temperature was 150 °C. The imaging area was selected according to the sample size, the spatial resolution was 50 μm × 50 μm and the spatial scanning rate was 100 μm/s. Imaging time was varied with area of the tissue samples. Imaging 1 cm × 1 cm area of tissue sample took approximately 6 h. 

### 4.5. Sampling for LC-MS/MS

The different organs/tissues of seedling at 30, 60, 90 day after planting were collected. The roots, stems and leaves of mature *G. elegans* were respectively divided into three parts according to the maturity degree, and the leaves were separated into mesophyll and vein. The sample preparation method was performed according to previously reported [54]. 100 mg sample was ground in a mortar with liquid nitrogen, and then extracted with 2.5 mL of 80% ethanol (*v*/*v*) in a 60 °C ultrasonic machine (Shumei KQ-250DE, Kunshan, China) for 30 min. The supernatant was collected by centrifugation equipment (Thermo Heraeus Megafuge 8R, Waltham, MA, USA) at 10,000× *g* for 10 min. This procedure was repeated twice and combined twice filtrates. A total of 1 mL of the filtered solution was evaporated by dry nitrogen, and then reconstituted with 200 μL 5 mM ammonium acetate-water (pH 5.2) and 800 μL acetonitrile solution. The supernatant was filtered through a 0.22 μm microbore cellulose membrane and injected via an autosampler vial for LC-MS/MS analysis.

### 4.6. Standard Curve Preparation for LC-MS/MS

A total of 0.01 g of standard chemical was accurately weighed and put into a small beaker. Reagent was ultrasonically dissolved in methanol, then diluted to 10 mL. The prepared standard solution (1.0 mg/mL) was stored at −80 °C in dark. The standard solution was diluted to 200 ng/mL, 100 ng/mL, 50 ng/mL, 20 ng/mL, 10 ng/mL, 1 ng/mL, and 0.1 ng/mL, respectively. The peak area and concentration were used as the ordinate and abscissa to draw the standard curve of 3 standard chemicals.

### 4.7. LC-MS/MS Instrumentation and Data Acquisition

The analytical method was performed according to previous report [54]. Alkaloid quantification in *G. elegans* was performed on an Agilent series 1290 Infinity HPLC instrument coupled with an Agilent 6460 MS/MS (Agilent Technologies, Santa Clara, CA, USA). The samples were separated on a Waters C18 column (150 mm × 4.6 μm i.d., 3.5 μm, Waters, Milford, MA, USA). The mobile phase consisted of solvent A (5 mM ammonium acetate-water, pH 5.2) and solvent B (acetonitrile solution), the flow rate was set to 0.3 mL/min, the column oven temperature was maintained at 40 °C, and the sample injection volume was 10 μL. The elution gradient program of the positive ion mode was as follows: 0–2 min, 10% B; 2–7 min, 10% B to 15% B; 7–20 min, 15% B to 35% B; 20–30 min, 35% B to 90% B; 30–33 min, 90% B; 33.01–40 min, 10% B.

The multiple reaction monitoring (MRM) was performed in the quantitative analysis of the target compounds. Nitrogen gas was used as the drying and collision gas. The MS/MS conditions were as follows: the flow rate of the nebulizer gas was 12 L/min, the capillary voltage was 3.50 kV, the nebulizer pressure was 40 psi, and the capillary temperature was 350 °C. Mass spectra were recorded over the mass range of *m*/*z* 50–1000. The data acquisition was controlled by the Agilent MassHunter workstation software (version B.07.00).

### 4.8. Imaging and Data Processing

The raw data collected by DESI-MSI were processed with the Masslynx software V4.1. The acquisition data were processed by the HDImaging 1.5 software to create the image files. In order to improve the clarity of images, optimal visualization parameters were selected. The image gradient was adjusted through image brightness intensity.

### 4.9. Statistical Analysis

All experiments were conducted in at least three replicates. Statistical significance was evaluated using Microsoft Excel software. Values were expressed as means ± SE.

## 5. Conclusions

In this study, *in situ* distribution of gelsemine, koumine, and gelsenicine in organs and tissues of *G. elegans* at different growth stages were visualizes by DESI-MSI. In order to validate the MSI method, LC-MS/MS were employed to accurately quantify the contents of three alkaloids in different parts of *G. elegans* plant. Through comprehensive analysis of the MSI images, gelsemine, koumine, and gelsenicine were found to mainly accumulate in plant pith and gradually reduce from center to edge in root and stem cross-section. Leaf vein contained higher level of three alkaloids comparing with mesophyll. Mature tissues of *G. elegans* were accumulated more gelsemine, koumine, and gelsenicine in root and stem. Synthetically, a high-resolution map about *in situ* distribution of gelsemine, koumine, and gelsenicine in *G. elegans* was generated. This study provided a useful protocol for the visualized *in situ* visual distribution of alkaloids in plants. These findings would contribute to well understand the accumulation in *G. elegans* organs/tissues at different growth stages and could guide the separation and purification of gelsemine, koumine, gelsenicine from *G. elegans*.

## Figures and Tables

**Figure 1 molecules-27-01810-f001:**
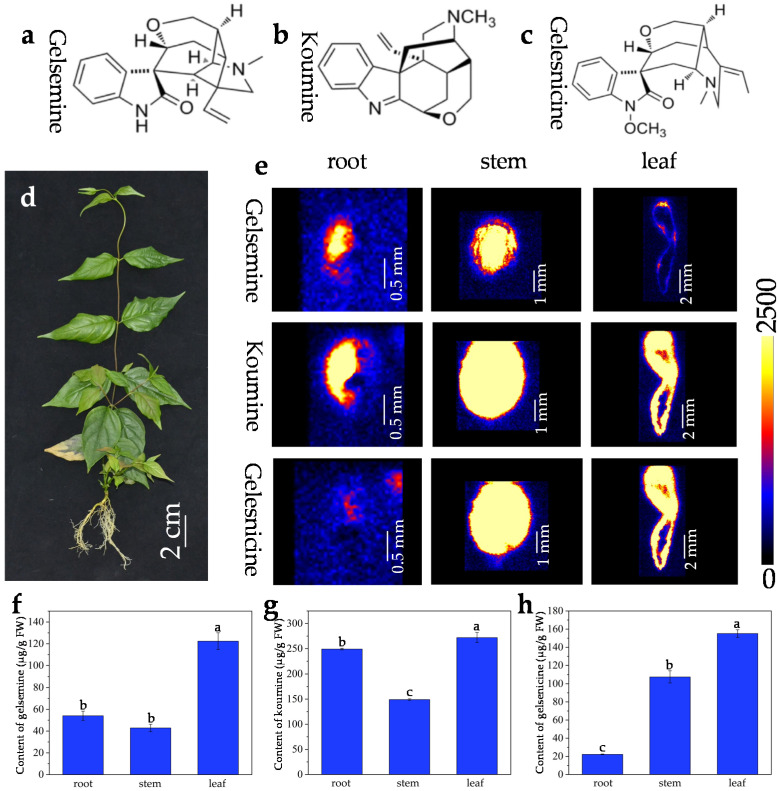
*G. elegans* plant and distribution of three alkaloids in mature organs/tissues. (**a**) Chemical structure of gelsemine (**a**), koumine (**b**), and gelsenicine (**c**); (**d**) mature *G. elegans* plant (about one year old); (**e**) *in situ* visualization of three alkaloids in mature *G. elegans* organs/tissues by DESI-MSI. The contents of gelsemine (**f**), koumine (**g**), and gelsenicine (**h**) in organs/tissues of mature *G. elegans*. The error bars indicate the standard deviations of three biological replicates, and statistical significance was evaluated using Microsoft Excel software. Different lowercase letter above the bars represented significance at the level of *p* < 0.05, and the same letter denotes a non-significant difference between the means.

**Figure 2 molecules-27-01810-f002:**
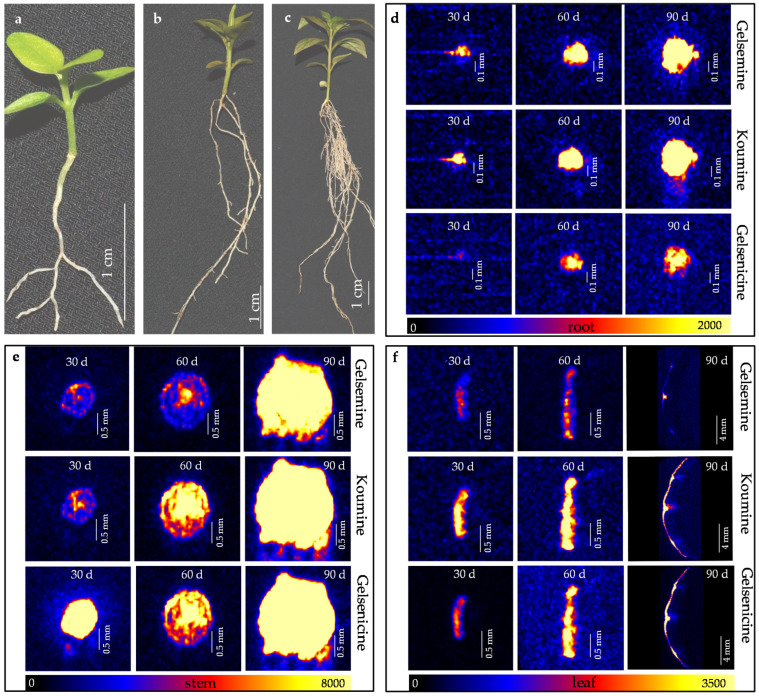
*In situ* visualization of three alkaloids in organs/tissues of *G. elegans* during the growth and development. 30 day (**a**), 60 day (**b**) and 90 day (**c**) old seedling of *G. elegans*; (**d**) The alkaloid visualization in root of 30, 60 and 90 day old *G. elegans* seedling respectively; (**e**) The alkaloid visualization in stem of *G. elegans* seedling at 30, 60 and 90 day; (**f**) The alkaloid visualization in leaf of 30, 60 and 90 day old *G. elegans* seedling respectively.

**Figure 3 molecules-27-01810-f003:**
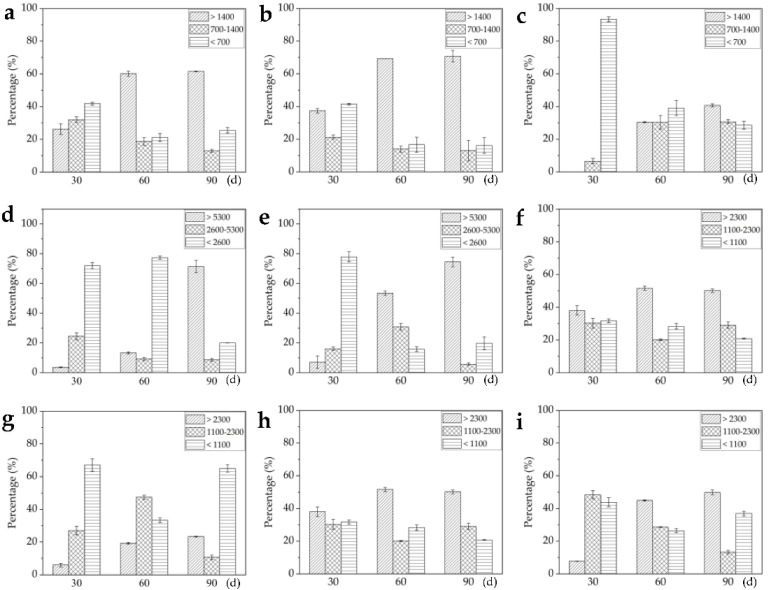
Area percentage evaluation of three alkaloids in plant tissues according to the image brightness intensities. (**a**–**c**) Area percentage of gelsemine, koumine, and gelsenicine in root in three intensity ranges (>1400, 700–1400, and <700); (**d**–**f**) area percentage of alkaloids in stem in three intensity ranges (>5300, 2600–5300, and <2600); (**g**–**i**) area percentage of alkaloids in leaf in three intensity ranges (>2300, 1100–2300, and <1100). The error bars indicate the standard deviations of three biological replicates. Statistical significance was evaluated using the Microsoft Excel software.

**Figure 4 molecules-27-01810-f004:**
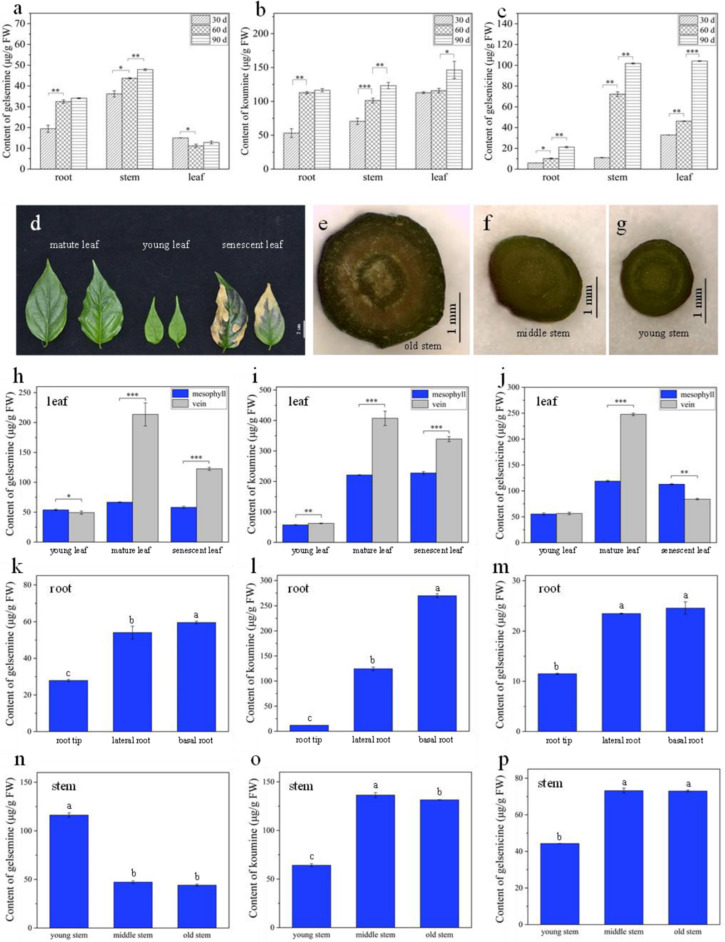
Quantification of three alkaloids in different parts of *G. elegans*. (**a**–**c**) The contents of three alkaloids in the roots, stems and leaves of 30, 60, and 90 day old *G. elegans*; (**d**) different old leaf of *G. elegans*; (**e**–**g**) cross-sections of different parts of *G. elegans* stem; (**h**–**j**) contents of gelsemine, koumine, and gelsenicine in different old parts of leaf; (**k**–**m**) contents of gelsemine, koumine, and gelsenicine in different old parts of root; (**n**–**p**) contents of gelsemine, koumine, and gelsenicine in different old parts of stem. The error bars indicate the standard deviations of three biological replicates, and asterisks represent statistically significant differences as determined by Fisher’s least significant difference (* *p* < 0.05, ** *p* < 0.01, *** *p* < 0.001). The presence of the same lowercase letter denotes a non-significant difference between the means (*p* < 0.05).

**Figure 5 molecules-27-01810-f005:**
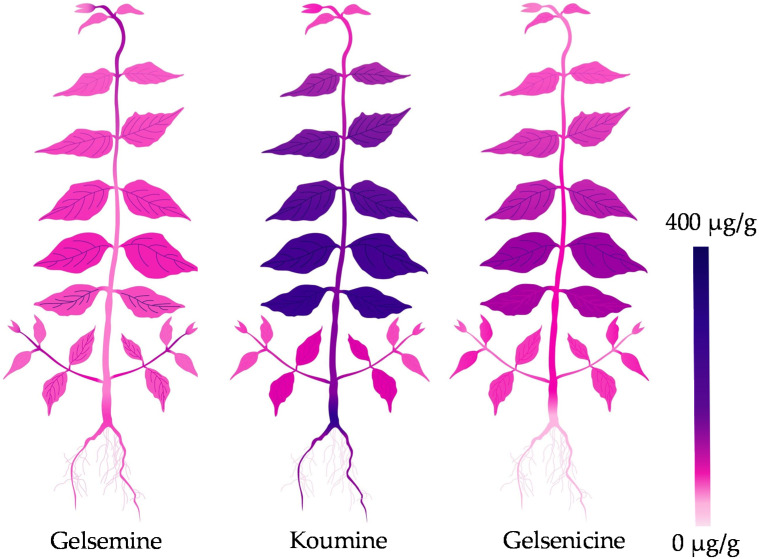
Diagrammatic distribution of three alkaloids in *G. elegans*.

**Table 1 molecules-27-01810-t001:** Area percentages (AP) of gelsemine, koumine, and gelsenicine in different intensity range. Statistical significance was evaluated using Microsoft Excel software. Values were expressed as means ± SE.

Tissue	Alkaloids	Area Percentages (AP, %)		
		>1400	700–1400	<700
Root	Gelsemine	49.0 ± 1.4	34.0 ± 2.5	17.0 ± 0.4
	Koumine	62.4 ± 0.6	21.1 ± 1.9	16.5 ± 1.2
	Gelsenicine	0.0	3.9 ± 1.8	96.1 ± 1.8
Stem	Gelsemine	61.0 ± 2.0	23.5 ± 3.1	15.5 ± 0.5
	Koumine	100	0.0	0.0
	Gelsenicine	100	0.0	0.0
Leaf	Gelsemine	4.9 ± 1.2	2.9 ± 3.1	92.2 ± 0.8
	Koumine	78.9 ± 0.2	17.1 ± 2.1	10.5 ± 1.9
	Gelsenicine	72.4 ± 2.2	16.8 ± 0.1	4.30 ± 1.0

## Data Availability

The data presented in this study are openly available for all figures and samples.

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
