# Peer review of "In Situ Visual Distribution of Gelsemine, Koumine, and Gelsenicine by MSI in Gelsemiumelegans at Different Growth Stages"

_molecules, 2022, doi:10.3390/molecules27061810_

Round 1
Reviewer 1 Report
In this manuscript entitled “in situ distribution of three alkaloids based on mass spectrometry imaging in Gelsemium elegans”, the authors have reported in situ distribution of three alkaloids in G. elegans plant by using DESI-MSI coupled with LC-MS/MS. Major revisions are needed to make the work acceptable.
- The title is informative and relevant, it could be more specific.
- Please start the abstract with a short introduction of the current problem(s) and the solution, based on the current study, in one or two lines.
- Please try to clarify why this is exciting work.
- The author should write at least 5 words as keywords. Also, authors should not use words that are already in the title.
- Why is no standard deviation of measurements visible in Figure 3? The authors should add the error bar to this figure.
- In the current state, there are some typographical errors. Therefore, the authors are advised to recheck the whole manuscript for improving the language and structure carefully.
- The results should be compared with data on other related work reported in the literature, otherwise, it is difficult to judge the actual impact/improvement beyond the state of the art of the present approach.
- In the “Introduction” section, general description on the importance of manuscript topic is poor. Therefore, the importance of this work cannot be well recognized from general readers. In order to fix this problem, addition of description on recent development in the field of research topic with citing recent comprehensive papers would be important. For improve “Introduction” part, related references should be cited: Composites Part B: Engineering 161 (2019) 564-577; Carbohydrate polymers 229 (2020) 115428; Ultrasonics sonochemistry 42 (2018) 171-182
Author Response
In this manuscript entitled “in situ distribution of three alkaloids based on mass spectrometry imaging in Gelsemium elegans”, the authors have reported in situ distribution of three alkaloids in G. elegans plant by using DESI-MSI coupled with LC-MS/MS. Major revisions are needed to make the work acceptable.
Response: Thank you very much for your valuable suggestions, which are very helpful for improving our manuscript. According to the comments, we carefully checked the text and have made corresponding revisions (tracked with red markings in the text). The revision details and item by item responses to the comments are as follows. We sincerely look forward to your approval or further comments.
- The title is informative and relevant, it could be more specific.
Response: Thank you very much. We added more information in title and retitled it as “in situ visual distribution of gelsemine, koumine and gelsenicine by MSI in Gelsemium elegans at different growth stages”. (Lines 2-3)
- Please start the abstract with a short introduction of the current problem(s) and the solution, based on the current study, in one or two lines.
Response: Thanks a lot for the valuable comment. We supplied a short introduction of the current problem(s) and the solution in Abstract. (Lines 12-19)
- Please try to clarify why this is exciting work.
Response: Thank you for your constructive suggestion. According to the comment, we rewrote the last paragraph of Introduction section and further highlighted the novelty and significance of our study. (Lines 80-89)
- The author should write at least 5 words as keywords. Also, authors should not use words that are already in the title.
Response: We agree with the comment, and adjusted keywords (including G. elegans, DESI-MSI, LC-MS/MS, distribution, alkaloid and visualization) according to the comment. (Line 29)
- Why is no standard deviation of measurements visible in Figure 3? The authors should add the error bar to this figure.
Response: Thank you for your reminding. We added the error bar in Figure 3 and Table 1 according to captured images of three adjacent frozen section. (Lines 170-174)
- In the current state, there are some typographical errors. Therefore, the authors are advised to recheck the whole manuscript for improving the language and structure carefully.
Response: Thank you for your comment, we rechecked and corrected the typesetting in revised manuscript. (Lines 32, 71, 137, 207, 225, 242-243)
- The results should be compared with data on other related work reported in the literature, otherwise, it is difficult to judge the actual impact/improvement beyond the state of the art of the present approach.
Response: Thank you very much for your valuable suggestion! We rewrote the Discussion section, and supplied more literatures and compared the results of our study with other studies. In the first paragraph, we introduced the application of MSI in analyte visualization and prominently discussed the differences in this study: We paid more attentions to the accumulation of three alkaloids in plant at different growth and development stages; We divided the plant tissues categories in more detail and analyzed the contents in various tissues; We creatively generated the diagrammatic map for the in situ distribution of three alkaloids in G. elegans according to the data detected by DESI-MSI and LC-MS/MS. In the second paragraph, we summarized that previous researches about G. elegans which mainly focused on identification, separation and quantification of medicinal ingredients. The creative work in our study is that we supplied more detailed analysis about spatiotemporal distribution and accurate content ranges of the three alkaloids in G. elegans organs/tissues. (Lines 245-295)
- In the “Introduction” section, general description on the importance of manuscript topic is poor. Therefore, the importance of this work cannot be well recognized from general readers. In order to fix this problem, addition of description on recent development in the field of research topic with citing recent comprehensive papers would be important. For improve “Introduction” part, related references should be cited: Composites Part B: Engineering 161 (2019) 564-577; Carbohydrate polymers 229 (2020) 115428; Ultrasonics sonochemistry 42 (2018) 171-182
Response: Thank you very much for the valuable suggestions and related references. We accordingly added more descriptions in the Introduction section and cited these works. (Lines 68-70)
Reviewer 2 Report
IIn general, the manuscript is well presented, but when quantifying a compound, it should present a standard error value since a previously validated method is being used. The values of the amounts obtained from the alkaloids in the different samples do not mention this standard error. The same happened with the values in table 1 and figure 3. This gives me to understand that the experiments were not done in triplicate and it causes me noise.
I recommend that the chemical structures of the mentioned alkaloids be added.
Author Response
Thank you very much for your valuable suggestions, which are very helpful for improving our manuscript. According to the comments, we carefully checked the text and have made corresponding revisions (tracked with red markings in the text). The revision details and item by item responses to the comments are as follows. We sincerely look forward to your approval or further comments.
- In general, the manuscript is well presented, but when quantifying a compound, it should present a standard error value since a previously validated method is being used. The values of the amounts obtained from the alkaloids in the different samples do not mention this standard error. The same happened with the values in table 1 and figure 3. This gives me to understand that the experiments were not done in triplicate and it causes me noise.
Response: Thank you very much for your positive comments and valuable suggestions. We added the error bar in Figure 3 and Table 1 according to captured images of three adjacent frozen section. (Lines 170-174)
- I recommend that the chemical structures of the mentioned alkaloids be added.
Response: We added the chemical structures of three alkaloids in Figure 1.
Reviewer 3 Report
Manuscript is well written but needs inclusion analytical method validation and some meaningful mass spectra of the alkaloids using DESI-MS and LC-MS/MS.

Author Response
Manuscript is well written but needs inclusion analytical method validation and some meaningful mass spectra of the alkaloids using DESI-MS and LC-MS/MS. Authors have applied the in situ mass spectrometry imaging system to visualize three alkaloids in Gelsemium elegans. The alkaloids have potential impact from medicinal point of view. The manuscript is well written, however, needs major revision.
Response: Thank you very much for your valuable suggestions, which are very helpful for improving our manuscript. According to the comments, we carefully checked the text and have made corresponding revisions (tracked with red markings in the text). The revision details and item by item responses to the comments are as follows. We sincerely look forward to your approval or further comments.
Major revision:
- Analytical method validation is required to justify the quantification of the three alkaloids, so it is strongly suggested to insert a sub-section of analytical method validation.
Response: Thank you very much. According to the comment, we supplied the method validation for LC-MS/MS in the sub-section of “2.4. Method validation” . Furthermore, we added the standard curve and mass spectra were presented in supplementary file (Lines 197-206). Considering this sub-section (2.4) is also a validation for MSI, thus we did not set another new sub-section.
- The manuscript is based on mass spectrometry analysis, so it is highly recommended to include some meaningful mass spectra in different Figure(s).
Response: Thank you for your reminding. We supplied the mass spectra and presented the figures in supplementary file.
Minor revision:
- A scheme of structures of the three alkaloids may insert in the text.
Response: We added the chemical structures of three alkaloids in Figure 1.
- Page 4, line 127, please check Figure number.
Response: We modified the figure number. (Line 137-139)
- Key words may need re-arrangement.
Response: We re-arranged keywords (including G. elegans, DESI-MSI, LC-MS/MS, distribution, alkaloid and visualization). (Line 29)
- Figure style should be same throughout the text.
Response: We unified the font format in figures.
- Page 9, line 300, it is better to use 4.50 kV instead of 4500 V (same as in page 10, line 337).
Response: Thank you very much. Accordingly, we replaced 4500 V and 3500 V with 4.50 kV and 3.5 kV. (Lines 343, 381)
- m/z is conventionally in italic, e.g., m/z.
Response: We corrected the font format for m/z in italic. (Lines 72, 342, 382)
Reviewer 4 Report
I think that the manuscript entitled “in situ distribution of three alkaloids based on mass spectrometry imaging in Gelsemium elegans" deserves publication in Molecules after major revision. The manuscript is very interesting, clearly presented results. However, it lacks some essential chapters: discussion, conclusion.
Line 32: why is family in italics
Line 120: what the letters above the bars mean
Line 123: please change „30d” into „30 d”
Lines 224-226: introduction
Line 269: how many batch of repeat plantings
Line 282: please complete information on the freezer incl. name, type, city, country
Lines 284-286: please complete information on the embedding machine incl. name, type, city, country
Line 310: please complete information on the ultrasonic machine incl. name, type, city, country
Line 311: please complete information on the centrifugation - equipment incl. name, type, city, country
Line 329: please complete information on the acetonitrile solution, which?
Complete lack of discussion of the obtained results with the literature data!
No description of the statistical methods.
No conclusion
Author Response
I think that the manuscript entitled “in situ distribution of three alkaloids based on mass spectrometry imaging in Gelsemium elegans" deserves publication in Molecules after major revision. The manuscript is very interesting, clearly presented results. However, it lacks some essential chapters: discussion, conclusion.
Response: Thank you very much for your positive comments and valuable suggestions, which are very helpful for improving our manuscript. According to the comments, we carefully checked the text and have made corresponding revisions (tracked with red markings in the text). The revision details and item by item responses to the comments are as follows. We sincerely look forward to your approval or further comments.
- Line 32: why is family in italics
Response: We corrected the format. (Line 32)
- Line 120: what the letters above the bars mean
Response: Thank you for reminding. Accordingly, we added a description for the letters above the bars in figure legend. (Lines 132-134, 242-243)
- Line 123: please change “30d” into “30 d”
Response: We corrected the error. (Line 137)
- Lines 224-226: introduction
Response: We rewrote the specified text in the Discussion section of the revised manuscript. (Lines 245-295)
- Line 269: how many batch of repeat plantings
Response: A total of 50 pots (each plot contained 2 plants) of plants were planted in this study, and we also supplied related description in Materials and Methods section. (Lines 314-315)
- Line 282: please complete information on the freezer incl. name, type, city, country
Response: Thank you for your reminding. Accordingly, we supplied the related information of freezer in Materials and Methods section. (Lines 324 and 334)
- Lines 284-286: please complete information on the embedding machine incl. name, type, city, country
Response: We supplied the information of the embedding machine. (Line 327)
- Line 310: please complete information on the ultrasonic machine incl. name, type, city, country
Response: We supplied the information of the ultrasonic machine. (Line 353)
- Line 311: please complete information on the centrifugation - equipment incl. name, type, city, country
Response: We supplied the information of the centrifugation - equipment. (Lines 354-355)
- Line 329: please complete information on the acetonitrile solution, which?
Response: We supplied the information of the acetonitrile solution. (Lines 301-302)
- Complete lack of discussion of the obtained results with the literature data!
Response: Thank you very much for your valuable suggestion! We rewrote the Discussion section, and supplied more literatures and compared the results of our study with other studies. In the first paragraph, we introduced the application of MSI in analyte visualization and prominently discussed the differences in this study: We paid more attentions to the accumulation of three alkaloids in plant at different satges in growth and development; We divided the plant tissues categories in more detail and analyzed the contents in various tissues; We creatively generated the diagrammatic map for the in situ distribution of three alkaloids in G. elegans according to the data detected by DESI-MSI and LC-MS/MS. In the second paragraph, we summarized that previous researches about G. elegans which mainly focused on identification, separation and quantification of medicinal ingredients. The creative work in our study is that we supplied more detailed analysis about spatiotemporal distribution and accurate content ranges of the three alkaloids in G. elegans organs/tissues. (Lines 245-295)
- No description of the statistical methods.
Response: Thanks a lot for the valuable comment. Accordingly, we inserted a sub-section of the statistical methods. (Lines 390-392)
- No conclusion
Response: We added the Conclusions section. (Line 393-407)
Round 2
Reviewer 1 Report
ACCEPT
Reviewer 4 Report
I think that the manuscript entitled “in situ distribution of three alkaloids based on mass spectrometry imaging in Gelsemium elegans" deserves publication in Molecules in present form.